# Assessment of Dietary Sodium, Potassium and Sodium-Potassium Ratio Intake by 72 h Dietary Recall and Comparison with a 24 h Urinary Sodium and Potassium Excretion in Dominican Adults

**DOI:** 10.3390/nu17030434

**Published:** 2025-01-24

**Authors:** Madeline Durán-Cabral, Rocío Estévez-Santiago, Alexandra Winter-Matos, Kilsaris García-Estrella, Begoña Olmedilla-Alonso, Carlos H. García-Lithgow

**Affiliations:** 1Dirección de Investigación, Universidad Nacional Pedro Henríquez Ureña (UNPHU), Santo Domingo 10602, Dominican Republic; madelineduranc@gmail.com; 2Facultad de Ciencias de la Salud, Universidad Francisco de Vitoria, Ctra. Pozuelo-Majadahonda Km 1800, 28223 Pozuelo de Alarcón, Spain; 3Centro Cardio-Neuro-Oftalmológico y Transplante (CECANOT), Santo Domingo 10306, Dominican Republiccgarcial@unphu.edu.do (C.H.G.-L.); 4Departamento de Metabolismo y Nutrición, Instituto de Ciencia y Tecnología de Alimentos y Nutrición (ICTAN-CSIC), 28040 Madrid, Spain

**Keywords:** salt intake, sodium, potassium, sodium-to-potassium ratio, dietary intake, Dominicans, Caribbean, Dominican Republic

## Abstract

**Background:** PAHO-WHO reports that sodium intake is currently high in the Caribbean. The objective was to estimate sodium (Na) and potassium (K) intakes by 72 h dietary recall and compare them with those obtained from 24 h urinary excretion in Dominican adults. **Methods:** A total of 69 adults (33 men) completed a 3-day dietary recall with emphasis on added salt and seasonings. The 24 h urine samples were analysed by indirect potentiometry using the membrane ion-selective electrode technique. The WHO-PAHO Questionnaire on Knowledge, Attitudes and Behaviour toward Dietary Salt and Health was completed. **Results:** Dietary Na intake ranged from 1.0 to 8.3 g. Median dietary and urinary Na concentrations were similar (2.7 and 2.5 mmol/d). Mean dietary Na and K concretertentrations were higher than those excreted in 24 h urine (133.0 ± 59.7 vs. 103.7 ± 44.5 mmol Na/d, *p* = 0.001; 69.0 ± 21.0 vs. 36 ± 16.3 mmol K/d, *p* < 0.001). The Na-to-K ratio was lower in dietary than in 24 h urine samples (2.0 ± 1.1 vs. 3.2 ± 1.6 mmol/d, *p* < 0.001). Urinary Na concentration was associated with sex (r = 0.280, *p* = 0.020) and obesity (r = 0.244, *p* = 0.043) and K with sex (r = 0.356, *p* = 0.003). Urinary Na-to-K was inversely related to age (r= −0.291, *p* = 0.015). Sex and obesity explained 11% of the variance in urinary Na concentration and sex only of the variance in urinary K concentration. The only significant correlation between dietary and urinary concentrations was that of K (r = 0.342, *p* = 0.004). This correlation matrix, controlled for overweight and sex, maintained the level of significance and was equal in almost 12% of the data. **Conclusions**: These data, which are the first data on Na and K intakes in Dominicans assessed by dietary assessment, showed a higher mean sodium intake (mean of dietary recall and urinary excretion data: 2.7 g Na, 6.8 g salt/day) and a lower K intake (2.06 g/day) than the WHO recommendations (<2.0 g Na, ≥3.5 g K). Potassium, but not sodium, intake from 72 h food recall and 24 h urinary excretion showed a correlation when controlling for sex and obesity, but not enough to consider them interchangeable.

## 1. Introduction

Excess salt intake is considered a major modifiable risk factor for chronic noncommunicable diseases, including cardiovascular disease (CVD) [1,2,3,4]. At the general public health level, the World Health Organization (WHO) recommends reducing sodium intake to less than 2 g per day (5 g salt/person/day) to lower blood pressure and reduce the risk of cardiovascular disease, stroke and ischaemic heart disease [5]. In addition, numerous dietary guidelines, such as the American Heart Association (AHA) [6,7] and American Diabetes Association (ADA) [8,9], provide sodium (Na) intake recommendations for the general population and also for at-risk subgroups (e.g., diabetics, hypertensive patients, people with chronic kidney disease). However, although they all recommend reducing sodium intake, there are discrepancies in the recommended amount, as, among other aspects, some research suggests that reducing sodium intake too much may be a risk for certain groups, such as type 2 diabetics [2]. On the other hand, the relationship between sodium intake and hypertension is not consistent across studies [10,11]. This relationship could be influenced by several factors [12], including other dietary components or characteristics of the populations studied (e.g., age, origin). Among the dietary components, potassium (K) stands out, as the dietary sodium-to-potassium (Na-to-K) ratio seems to be more relevant than simple sodium intake when assessing the impact on blood pressure and cardiovascular disease morbidity and mortality [12,13,14,15], as it has been proposed as an independent predictor of stroke risk [16].

The WHO recommends reducing sodium intake and increasing dietary potassium intake, as insufficient intake (less than 3.5 g potassium/day), together with high sodium intake, contributes to increased blood pressure and the risk of coronary heart disease and stroke [17,18]. In the same vein, the member countries of the Central American Integration System (SICA, for its Spanish acronym) have developed policies and plans to prevent and control noncommunicable diseases (NCDs), combat overweight and obesity, and reduce salt/sodium intake in the population (reducing salt/sodium intake in the population by 30% is one of the targets of the Global NCD Action Plan for 2025). In 2018, the Dominican Republic approved the Regional Strategy for Salt and Sodium Reduction in Central America and the Dominican Republic 2019–2025 [4].

According to the Pan American Health Organization (PAHO/WHO) [19], current salt, and therefore sodium, consumption in the Caribbean region far exceeds WHO recommendations, ranging from 8.5 to 15 g of salt (3.4 to 6 g of sodium) per person per day. However, although PAHO had planned to conduct dietary surveys on salt intake in the Dominican Republic in 2019 [20], we are not aware that they have been carried out, but it is necessary to know before implementing strategies to reduce salt consumption. The assessment of salt intake through dietary surveys is the most widely used method at the population level, although the gold standard method is the determination of 24 h urinary sodium excretion [21]. To the best of our knowledge, the only data on sodium intake in Dominican subjects are those recently published by our group [22] on 24 h urinary sodium and potassium excretion in Dominican adults, with mean sodium and salt intakes (2.3 and 5.8 g/day, respectively) slightly higher and potassium intake (1.4 g/day) lower than those recommended by the WHO. The aim of the present study was to estimate sodium and potassium intakes using the 3-day food recall in a subset of this study and to compare these data with those obtained from the assessment of sodium and potassium excretion in 24 h urine samples.

## 2. Materials and Methods

### 2.1. Subjects and Study Design

A total of 69 adults (33 men, 36 women) who completed a 3-day dietary recall were selected from participants in a previously published study to assess salt intake by measuring sodium and potassium in 24 h urine samples [22]. The majority were overweight/obese (75.4% with a body mass index (BMI) > 25–32). Volunteers were asked to provide information on the following exclusion criteria: diabetes or nephropathy with complications (e.g., renal, ocular), chronic diseases, pregnancy and use of restricted diets or avoidance of any food group.

The study was conducted in accordance with the tenets of the Declaration of Helsinki, and all procedures involving human subjects were approved by the Consejo Nacional de Bioética en Salud (CONABIOS) (Registry No. 022-2020, dated 9 December 2020). In the Dominican Republic, the identity of participants was protected during the handling of samples and data, in accordance with Law No. 172-13, G.O. and No. 10737 of 15 December 2013. Written informed consent was obtained from all subjects.

Body weight (kg) was measured without shoes and in light clothing. Height was recorded to the nearest cm using a scale (8023 Jiangsu Medical SH, Jiangsu Scale). BMI was calculated as weight (kg)/height (m^2^).

### 2.2. Dietary Intake Assessment

Three-day food recalls were used to calculate dietary intakes of sodium, potassium, macronutrients (protein, lipids and carbohydrates) and calories. For the first recall, participants had a face-to-face meeting with a specialist interviewer, usually the same person who then carried out the other two recalls by telephone. These recalls took place over a seven-day period, one of which coincided with a weekend or holiday. In the dietary recalls, particular emphasis was placed on questions about salt intake, with participants being asked about the amount of salt added at the table and during cooking, food seasonings (known by Dominicans as “sopita”, a product similar to chicken stock cubes) and seasonings/condiments (powder or liquid, known by Dominicans as “sazonador”), as well as the trade name of the brand used. In addition, a pinch of ’salt’ was considered to weigh 0.45 g, and this was the amount added to the following foods at the time of the food recall: avocado, butter, nuts (a handful) and ‘sazón’ (if reported by the participant).

Amounts consumed were estimated in units (fruit), portions or household servings (based on food weighed by us or widely used household measuring charts) [23,24]. Daily intakes of macronutrients, sodium and potassium were calculated in grams/day using the DIAL^©^ software, version 3.15 [25], which includes a food composition table (FCT) that was expanded for this study to include Dominican foods and recipes. The composition of these foods was extracted mainly from food composition tables, which include central and Iberoamerican foods [26], from the nutritional label supplied by the manufacturer and a few data from North American FCT [27]. The recipes were mainly taken from a classic and widely used Dominican recipe book [28] and a few others from another source consulted by the Dominicans [29]. The assumptions made for foods to be included in the DIAL programme were as follows (the name in the DIAL in brackets): grilled meat (fatty meat), griddle meat (semi-fatty meat), guava juice (tropical juice), “pipian” (stewed pork) and homemade orange juice was distinguished from commercial orange juice.

To complement the information on dietary intake, a WHO-PAHO questionnaire on knowledge, attitudes and behaviours towards dietary salt [30] was administered to the initial total sample (n = 149), consisting of eight questions to assess the level of knowledge about salt intake recommendations and the consequences of exceeding them, as well as current consumption habits and the intention to improve them.

### 2.3. Analysis of Sodium and Potassium in Urine Samples

The collection and analysis of 24 h urine samples have been described (urine completeness was determined by 24 h urine creatinine analysis and by the volume collected) and published elsewhere [22]. Briefly, urinary sodium and potassium excretion was analysed by indirect potentiometry using the membrane ion-selective electrode technique with the Beckman Coulter AU5811 analyser (Beckman Coulter Inc., Brea, CA, USA).

### 2.4. Statistical Analysis

Data are expressed as mean, standard deviation and range. The normal distribution of the data was assessed (Kolmogorov–Smirnov test). If it is not normally distributed, a new variable is generated by log+1 transformation. *T*-test and Mann–Whitney were used to compare data with normal distribution (or after logarithmic transformation) or when data were not normally distributed.

Correlations between dietary sodium and potassium intake and variables of sex, age, overweight, calorie and macronutrient intake were established using Spearman’s rho correlation coefficient for variables without normal distribution and Pearson’s coefficient for variables with normal distribution or after logarithmic transformation.

Multivariate regression analyses were performed using stepwise and intro models. Dietary sodium or potassium intake was a dependent variable, and sex, weight, age, calorie, protein, lipid and carbohydrate intake were independent variables.

With the aim of identifying multivariate relationships using the correlation matrix, a factorial analysis was performed (extraction method: principal component analysis; rotation method: Varimax rotation with Kaiser normalisation) to examine the following set of variables: protein, lipids, carbohydrates, calorie intake and overweight, sex and age. The results are presented in the form of component loading plots. Each point is connected to the origin, and the angles between the segments express the measure of the correlation (angles smaller than 90 ° indicate a positive correlation, and larger angles indicate a negative correlation).

All reported *p*-values are based on a two-tailed test, and a *p*-value < 0.05 was considered to indicate statistical significance. IBM^®^ SPSS^®^ Statistics for Windows, version 29.0, was used for all statistical calculations.

## 3. Results

The Dominican recipes and foods reported as consumed by the participants in this study were included in the food composition table of the DIAL© software, version 3.15 [25] and used for the analysis of dietary sodium, potassium and macronutrient intakes. The recipes are shown in Table 1, and the foods whose composition was obtained from manufacturer, supermarket or app nutritional labelling are shown in Appendix A. Foods whose composition was not available in the DIAL software and which were obtained mainly from the Instituto de Nutrición de Centro América y Panamá (INCAP) [26] and to a lesser extent from the United States Department of Agriculture (USDA) [27], are listed in Appendix A.

The dietary intake of the study participants was characterised by the consumption of rice (93%), natural juices (72.5%, mainly orange juice and to a lesser extent lemon, strawberry and tamarind juices), pulses (66%, mainly red beans, black beans and “guandules” (*Cajanus cajan*)) and vegetables (62%, mainly lettuce, tomato, cabbage, broccoli, aubergine and maize). On the other hand, consumption of fresh fish (not canned) and fruit is low, with 81% not consuming fish and 58% not consuming fruit. The fruit consumed by almost all participants was ripe bananas (“guineo”). Other fruits consumed were melon, pineapple, strawberries and ‘golden apples’. A number of starchy foods (grouped as ‘víveres’) are widely consumed, including plantain (“plátano macho”: *Musa Paradisiaca*, consumed raw and cooked), cassava (yucca), auyama, yautía (white, purple, coconut), sweet potato, yam and mapuey. Street food was consumed by half of the participants (51%): pizza (the majority), hamburgers, “pastelitos”, quipes, chicken flute and cassava dumplings.

During the three-day dietary recall, particular attention was paid to the issue of salt added to food as a pinch of salt, as a seasoning (“sazonador”) and as a food seasoning (“sopitas”). A pinch of salt was considered to be, on average, 0.45 g [23,31]. A common practice among the volunteers was to add seasoning in addition to salt when preparing meals. The sodium content of the food seasoning (‘sopitas’ = concentrated broth) was taken as the average (21.9 g/100 g) of the values provided by the INCAP tables of food composition [26], as it was considered representative of this food seasoning in Central America (24.0 g/100 g) and the values provided by the commercial brands most used by the participants (Doña Gallina: 18.9 g/100 g and Maggi: 22.7 g/100 g). The sodium content of the seasoning was 23.2 g/100 g (average of the values reported by INCAP (18.6 g/100 g), and the composition declared on the label of the brands Ranchero (31.1 g/100 g) and Maggi (20.0 g/100 g). Based on the data provided by the commercial brands, the average weight of seasoning (‘sopita’) and seasoning (‘sazonador’) added per portion was 2.2 g (0.5 mg sodium) and 3.8 g (0.9 mg sodium), respectively.

Dietary intakes of macronutrients (proteins, lipids and carbohydrates) and calories are shown in Table 2. Proteins provided 15% of their calories, total fat 45% and carbohydrates 40%. Sodium, potassium and the sodium-to-potassium ratio in the diet and in the 24 h urine samples for the whole sample and grouped by sex (52.2% women) are shown in Table 2. The dietary sodium concentration was higher than that excreted in the 24 h urine sample (133.0 ± 59.7 mmol/d vs. 103.7 ± 44.5 mmol/d, *p* = 0.001), as was the potassium concentration (69.0 ± 21.0 vs. 36.0 ± 16.3 mmol/d, *p* < 0.001). In contrast, the Na-to-K ratio was lower in the diet than in the 24 h urine samples (2.0 ± 1.1 vs. 3.2 ± 1.6 mmol/d, *p* < 0.001). In the diet, men had higher concentrations of potassium, calories, proteins and lipids than women. In the urine samples, both potassium and sodium were higher in men than in women. Most of the participants were overweight/obese (75.4%), and when comparing the dietary and urinary variables, there was only a significant difference in urinary sodium, which was higher in the obese (109.8 ± 40.0 [40.0–202.0] mmol/d) than in the normal weight subjects (84.8 ± 52.9 [23.0–191.0] mmol/d).

Dietary sodium and potassium intake data assessed by urinary excretion showed a normal distribution but not those assessed by dietary recall. The sodium centiles by dietary assessment were 3.6 g, 2.7 g and 2.2 g Na for the 75%, 50% and 25% centiles, respectively, and by urinary excretion: 2.9, 2.5 and 1.5 g Na, respectively. The potassium centiles by dietary assessment were 3.2 g, 2.6 g and 2.1 g K for the 75%, 50% and 25% centiles, respectively, and by urinary excretion, 1.9, 1.4 and 0.8 g, respectively.

### 3.1. Dietary Intake of Na and K in Relation to Macronutrient Intake, Sex, Age and BMI

Na and K intakes were correlated with protein, fat and calorie intakes (Table 3). Strong correlations are shown for K intake, which also correlates with carbohydrate intake and with sex (higher in men). However, the Na-K ratio did not show a significant correlation with any of the dietary variables. Sex correlated with calorie intake (0.435, *p* = 0.000), protein intake (0.386, *p* = 0.001) and lipid intake (0.362, *p* = 0.002). Age showed no statistically significant correlations. Weight was inversely correlated with protein intake (−0.243, *p* = 0.045).

Table 4 shows the statistically significant results of a multivariate regression analysis used to assess the predictive value of calorie, protein, lipid and carbohydrate intake, sex, overweight and age on dietary Na and K intakes. The R^2^ for Na intake was 0.162 with a significant direct effect of calories and overweight. However, for the K intake, a higher proportion of the variance, 48% (R^2^ = 0.476), was explained by the direct effect of calorie and protein intake and the inverse effect of lipid intake.

The correlation matrices for sex, age, overweight, calories, proteins, lipids, carbohydrates and Na and K intakes are shown in Figure 1 and Figure 2. The model explains 56.2% of the variance in Na intake related to calories, proteins, lipids and carbohydrates and, to a lesser extent, to overweight and sex (Figure 1). The model explains 59.4% of the variance in K intake, which is also related to calories, proteins, lipids and carbohydrates, but to a lesser extent to sex and age (Figure 2). The model is not statistically significant for the Na-K ratio.

### 3.2. Na and K in 24 h Urine Samples in Relation to Sex, Age and BMI

Na concentration in the urine shows a significant association with sex (r = 0.280, *p* = 0.020) and overweight (r = 0.244, *p* = 0.043) (men > women, overweight > normal weight). Urinary K is related to sex (r = 0.356, *p* = 0.003; men > women). Urinary Na-to-K is inversely related to age (r = −0.291, *p* = 0.015).

In a regression model, 11% of the variance in Na and K urine concentrations is explained by sex and overweight for Na urine and only by sex for K urine concentrations. Age and overweight explained 12% of the variance in the Na-K ratio.

### 3.3. Correlations Between Dietary Na and K Concentrations and 24 h Urine Samples

The only significant correlation between dietary and urinary concentrations was that of potassium (r = 0.342, *p* = 0.004). This correlation matrix, controlled for overweight (r = 0.356, *p* = 0.003) and sex (r = 0.267, *p* = 0.028), maintained the level of significance (R^2^ = 0.117) and was equivalent in almost 12% of the data (r^2^ = 0.117).

The correlations for sodium intake and for the salt intake (estimated by dietary assessment and urinary sodium excretion) were not statistically significant (r = 0.099 and p = 0.419, r = 0.190 and *p* = 0.117, respectively), nor was the relationship for the Na-to-K ratio (r = 0.214 *p* = 0.077). Neither were the correlations of the sodium grouping into men and women statistically significant. Overweight did not affect the correlations between Na and the Na-to-K ratio.

### 3.4. Questionnaire on Knowledge, Attitudes and Behaviour Towards Dietary Salt

This questionnaire [30] was completed by 149 volunteers. When asked about their salting habits at the table, 77.2% said they never or rarely added salt at the table, 53.0% said they never added salt, and only 3.4% said they always added salt. This response is consistent with the fact that 83% of respondents said that salt is always added when preparing food at home.

With regard to their perception of their current salt intake, it is worth noting that 50% of respondents consider their salt intake to be ‘just right’, and 35% consider their salt intake to be too low. This is not consistent with the intakes measured by two methods in this study, so there is an underestimation of perceived salt intake.

Almost 100% of respondents said they were aware of the adverse health effects of high salt intake, the most common being high blood pressure (84% of respondents), followed by kidney stones (14%). This knowledge is in line with the attitudes of 83% of respondents, who say that it is very important for them to reduce salt or sodium in their diet, and 88% of them say that they take measures to do so, such as avoiding or minimising the consumption of processed foods (56%) and other less popular measures such as checking food labels for salt or sodium content or not adding salt at the table (16% and 17% of the sample, respectively).

## 4. Discussion

The results of this study on sodium and potassium intake, together with those recently published on their excretion in 24 h urine samples [22], are the first data available on Na and K intake in Dominicans, despite the increasing prevalence of hypertension in this country (33.7% in 2022) [31,32] and overweight/obesity (70.8% in 2022) [32], as well as the great public health interest in these risk factors for hypertension, CVD and kidney disease [4,32]. Although the Dominican Republic belongs to the Caribbean and Central America, geographical areas for which there are studies estimating sodium/salt and potassium intakes [19,33,34], these studies do not include data obtained in the Dominican population, which, despite similarities in the use of ingredients (e.g., rice, beans and bananas), has its own dietary habits in terms of food preparation and seasoning. In order to ensure a better representation of participants’ usual consumption patterns, this study used a 72 h food recall rather than the 24 h recall generally used by government agencies in national surveys [30].

When comparing dietary intakes of macronutrients and calories with the nutrient targets proposed by PAHO for the Caribbean population [35], the mean calorie intake was similar (2250 kcal/day proposed by PAHO vs. 2333 kcal/d in the present study), but with a very wide range for calorie intake. According to these nutrient intake targets, the percentage of total calories from carbohydrates should be 65%, from total fat 25% and from protein 10% [35]. Instead, in this study, carbohydrates contributed a lower percentage (40%), and fats and proteins had a higher percentage (45% and 15%, respectively). These percentages are consistent with those described in a study of overweight and obese adult Dominicans (n = 50), in which the percentage contribution of carbohydrates was also 40%, but that of proteins was higher (20%), and that of fats was lower (40%) [36]. Therefore, the participants in this study should benefit from the recommendations included in the new nutrient targets for the Caribbean population, which suggest the need to increase the consumption of fruits, vegetables, legumes and nuts in the context of the total diet [35]. In this study, the consumption of fruit and vegetables was lower than recommended (over half ate no fruit and a third no vegetables) and with a very limited variety of fruit and vegetables (see Table 1), being mainly plantains (Musa paradisiaca) despite the diversity of the country [28].

In this study, the average daily amount of sodium and potassium was assessed by dietary recall (3.1 g Na, 2.7 g K) and by 24 h urine collection were different, with the concentration being higher when intake was assessed by the 72 h dietary recall. Instead, looking at the median concentration, sodium concentrations analysed by 72 h dietary recall and in 24 h urinary excretion are quite similar (2.7 g/d vs. 2.5 g/d) but not the median potassium concentration (2.6 g/d vs. 1.4 g/d). On the contrary, studies around the world have shown that sodium intake is underestimated when assessed by 24 h dietary recall versus 24 h urine collection [37,38,39,40]. In this study, a 3-day dietary recall was used instead of a 24 h dietary recall, which would be expected to result in less underestimation. However, the high sodium result from the 72 h dietary recall could also be influenced by an overestimation of the consumption of high-sodium foods added during the preparation of the recipes. Secondly, there is a slight overestimation of the fixed amount of salt in each of the dishes consumed by the participants, whereas in reality, the salt added to a given meal varies from day to day. On the other hand, although the gold standard for determining dietary salt intake is sodium analysis in a 24 h urine collection [21,41], a single 24 h urine sample at intakes in the range of 6 to 12 g salt/day was not suitable for detecting a 3 g difference in individual salt intake, as the steady state between sodium intake and excretion appears to require more than one, and even more than several, days to be reached [42]. In addition, sodium and potassium losses through sweat and faeces were not taken into account for a more accurate estimate of their excretion. In sweat, sodium losses can be higher than 10% depending on the ambient temperature, and during the study, the mean temperature was 32 °C, and the relative humidity was 86% [22]. Using this percentage of urinary sodium losses, the mean dietary sodium intake was 2.7 g/day (equivalent to 6.8 g salt intake/day). On the contrary, it is difficult to explain the difference of almost twofold between the K concentration estimated by dietary recalls and that obtained by urinary excretion, which could be due on the one hand to the fact that urinary excretion of potassium is considered an uncertain indicator of potassium intake [43] and on the other hand to the wide data variability in the K content in fruits and vegetables in the FCT used in this study, which is linked to the numerous variables that influence in the concentration of this mineral in these fruits (e.g., growing conditions, soil geochemistry and fertiliser use), major contributors to its intake. In this study, the sodium concentration in 24 h urine samples (mean 2.4 g/d) is lower than that in 24 h urine samples from South America and the Caribbean (several studies, mainly from Brazil), where the estimated sodium consumption over the last decade was 3.8 g/day (CI95%: 3.3–4.4 g/d) [34]. This is also lower than the 3.6 g sodium/day reported in a representative sample of Thai adults [44]. However, the present data on urinary Na concentration (median 2.5 g Na) are similar to those described in a Malaysian study (median 2.6 g Na) with a similar sample size of subjects [45] living in a comparable climate and with a comparable percentage of overweight, sex ratio and age range. It is also similar to that of a representative sample of people from Barbados in the Caribbean (2.7 mg sodium/d) [33].

In this study, the range of sodium intake was between 1.0 and 8.3 g, which is wider than that described in other Caribbean and Central American populations, which ranged from 3.4 to 6.0 g/person/day [19]. Furthermore, only 25% of the participants had a sodium intake (2.2 g Na/d from diet and 1.5 g Na/d from urinary excretion) close to that recommended by the WHO (≤2.0 g/day). Therefore, the recommendation to reduce salt intake would apply to most of the participants. Regarding potassium intake, about 90% of the participants had an intake below the recommended amount (≥3.5 g/d), which would benefit from achieving the recent PAHO target for the Caribbean population to increase fruit and vegetable consumption [35]. Fruit and vegetables are important sources of potassium and their intake was low in this study, as it was in another group of overweight and obese Dominicans, where it was below the recommendations for Dominicans [36,46]. Furthermore, according to a recent study on dietary intake in the Dominican Republic, as in the present study, the variety of fruits and vegetables most commonly consumed is very limited [47], with the five most important products in the Dominican diet being fresh chicken, rice, purified water, green plantain (always consumed cooked) and salami [47]. According to some studies, the consumption of fruit and vegetables is being replaced by ultra-processed foods, which ultimately affects the correct Na and K balance [4,19].

The molar ratio of Na-to-K was lower when calculated from dietary recall (2.0 ± 1.1 mmol/d) than from urinary excretion (3.2 ± 1.6 mmol/d), although in both cases, it was higher than the WHO recommended value [5]. In this study, the Na-to-K ratio using data from urinary excretion was higher than that found in adults from Barbados in the Caribbean (2.0) and also showed an inverse association with age [33] and an indirect association with overweight, as reported in this region.

Sodium and potassium intakes are correlated with intake of the three macronutrients, calories and overweight, with stronger associations for K than for sodium (47.6% vs. 16.2%). Dietary sodium intake did not differ by sex, but when urinary Na and K excretion was assessed, it was higher in men than in women, as recently described in a series of studies from more than fifty countries (in the WHO European Region) in which Na intake was assessed using different methods [48], in a representative Malaysian survey [49] and in a representative sample from Barbados [33]. In addition, higher urinary Na excretion was associated with overweight in this and other studies [48,49], which could be explained by the consumption of a greater amount of energy-dense foods that increase sodium in the daily diet [49]. For example, according to household surveys on living conditions in the Dominican Republic, salt consumption from food purchased in a fortnightly period exceeds ten grams per day [26]. Other studies have shown that not only urinary Na excretion but also urinary K excretion is associated with BMI [50].

Both mean sodium and potassium intakes, assessed by dietary recall or urinary excretion, differed from recommended amounts (<2.0 g sodium/d, ≥3.5 g potassium/d) [5], being higher for sodium (3.1 g/d by dietary recall and 2.4 g/d by urinary excretion vs. <2.0 g/d) and lower for potassium (2.7 g by dietary recall and 1.4 g by urinary excretion vs. ≥3.5 g/d). The Na-to-K ratio was two to three times higher than desirable, directly related to obesity and inversely related to age.

There was no correlation between sodium data from food recalls and 24 h urinary excretion, even when adjusted for sex and overweight, possibly because of the large number of factors that influence them and are not usually controlled for [40]. The range of dietary Na concentrations is much wider than that of urinary Na, about twice that of urinary Na. For potassium, although there is a correlation between measured dietary and urinary concentrations (12%), this value is not sufficient to consider the two types of measurements as “interchangeable” when controlling for sex and obesity. Therefore, given the complexity of the issues and the factors influencing the determination of dietary and urinary excretion, it may be appropriate to combine both data in studies, where available, to improve accuracy, as others have suggested, while improving existing methodologies [45].

Relevant strengths of this study are the detailed identification of the sources/recipients of the Dominican cuisine, the careful identification of its composition using appropriate FCT, the emphasis on questions about salt intake (salt added at the table and during cooking, food seasonings, etc.) and the use of a 72 h food recall, instead of the 24 h recall generally used in most studies, to better represent the usual consumption patterns of the participants. However, there are also limitations, such as the fact that this sample is a subset of a previous study in which the sample size was calculated to test for differences between normotensive and hypertensive subjects [22], so the comparison between dietary and 24 h urinary excretion data was performed on a sample size that was not calculated for this purpose. In addition, the Dominican participants are not representative of the Dominican population and, therefore, the results cannot be extrapolated to a population level.

## 5. Conclusions

Finally, several conclusions can be drawn from this work in adult Dominicans, in which sodium and potassium intakes were assessed from three 24 h food recalls, with particular emphasis on the amount of salt added to the food, and these data were compared with the corresponding 24 h urinary excretion. These data on sodium, potassium and salt contribute to filling the information gap on salt intake in Dominican subjects because this assessment of food recalls, together with the previously published study [22], measured by 24 h urinary excretion, are the only ones available in this population, whose culinary diversity differs from the gastronomy of other Caribbean countries, where several studies have been carried out.

In this sample, the mean sodium and potassium concentrations determined by dietary intake (3.1 g Na, 2.7 g K) were higher than those determined by urinary excretion (2.4 g Na, 1.4 g K). The mean salt intake obtained from the two methods was 6.8 g/day (dietary intake and urinary excretion were 7.7 and 5.9 g/day, respectively). With both methods, sodium intake is higher than the WHO recommendation (<2.0 g Na/day) [4] in more than 75% of the study participants. In addition, the majority (90%) had potassium intakes below WHO recommendations (≥3.5 g K/day), so they would benefit from the recent PAHO target for the Caribbean population to increase fruit and vegetable consumption [35]. On the other hand, only sodium and potassium intake assessed by urinary excretion, but not by dietary intake, were significantly associated with sex (Na and K) and with obesity (sodium). Age, sex and obesity influence the Na-to-K ratio, which is higher than desirable. Sodium intake data from 72 h food recall and 24 h urinary excretion showed no association, even when adjusted for sex and overweight. However, potassium intake data from food recall and urinary excretion showed a significant correlation when adjusted for sex and obesity, but not enough to be considered interchangeable.

## Figures and Tables

**Figure 1 nutrients-17-00434-f001:**
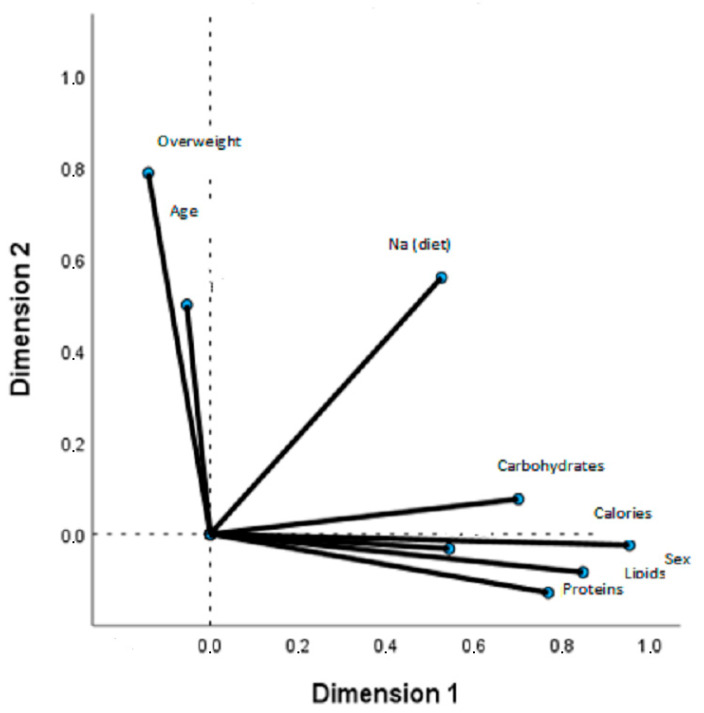
Factor analysis loading plot. Total group. Two principal components. Dietary Na and carbohydrates, proteins, lipids, calories, sex, age and overweight.

**Figure 2 nutrients-17-00434-f002:**
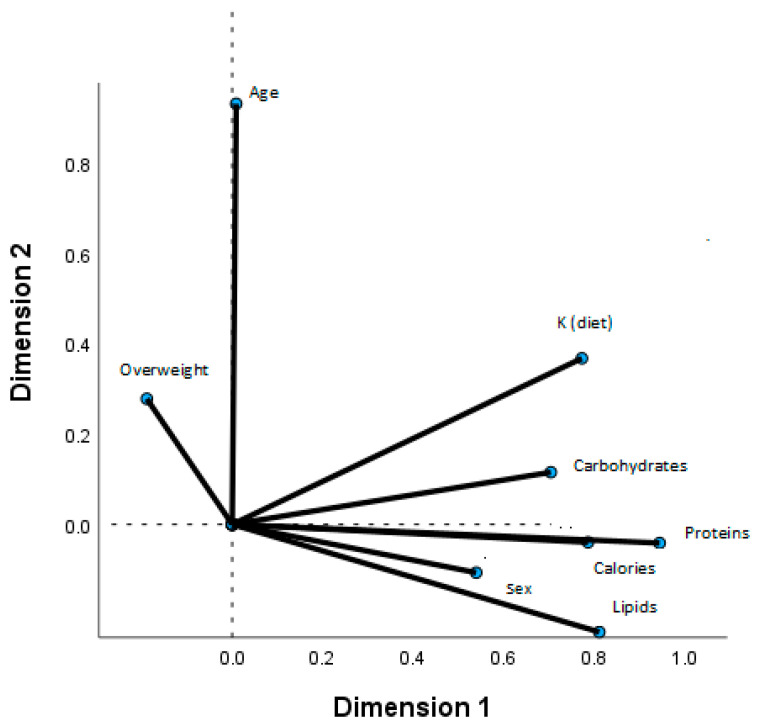
Factor analysis loading plot. Total group. Two principal components. Dietary K and carbohydrates, proteins, lipids, calories, sex, age and overweight.

**Table 1 nutrients-17-00434-t001:** Dominican recipes consumed by the participants in the study (“Spanish name”) and their bibliographical sources.

Aubergine pie with cheese [28]
Cake—three three milks [28]
Cassava (“yuca”, “arepitas de yuca”) [28]
Cassava pie (“catibía”) [28]
Cassava—mashed [29]
Chicken/cutlet (“asopao”: stew that can be prepared with chicken, pork, beef, shrimp, seafood, vegetables or any combination of these) [28]
Chicken potato cake [28]
Chicken—sweet and sour [28]
Coconut—baked sweet [28]
Cordon Bleu breasts [28]
Corn (“arepa”: corn flour “torta”) with coconut [28]
Cornmeal buns [28]
Cow leg stew [28]
Dominican Chowfan (fried rice with or without cooked chicken, cutlets or pork or shrimp dough) [28]
Dominican dumplings [28]
Dominican red pasta [28]
Flour (“El negrito”^®^) [28]
“Kibbe” (fried mix of minced meat, bulgur and spices) [28]
“Locrio” (rice stew with tomato sauce and meat, fish, offal, vegetables, etc.) [28]
Majarete (custard-like dessert made with shredded corn, cornstarch and coconut milk) [29]
Malfouf (wrapped cabbage rolls) [28]
“Mangú” (mashed green or ripe plantains, water, salt and butter, served with onions fried in oil and vinegar, salami, fried white cheese or avocado) [28]
Meatballs—stewed [28]
“Mofongo”—roasted and mashed green plantain with pork crackling and broth [29]
“Moro de güandules”—“Moro” = rice with pulses (red, black or “guandules” beans) [28]
“Moro de habichuelas”—beans and rice [28]
Octopus—grilled [Revista Semanal HOLA—cocina. Available online: https://www.hola.com/cocina/ (accessed on 18 December 2024)]
“Pico de gallo” (dressing made with tomato, onion, green chilli, coriander, lemon and olive oil [29]
Peanuts—sweet ground [28]
Plantain—fried ripe [28]
Pork stew [28]
Rice with noodles [28]
Rice with corn [28]
“Sancocho” (broth made from beef, pork, chicken or hen and vegetable products such as potato, yucca, yautía, auyama and green plantain) [28]
Soursop juice (“guanábana”) [28]
Vegetable cake/vegetable soufflé [29]
“Yaniqueque” (fried “torta” made with wheat flour, water and oil, which is eaten with sugar or salt and can be filled with cheese, ham or hard-boiled egg) [28]
“Yaroa” = fried potatoes or ripe plantains with melted yellow cheddar cheese and minced beef, garnished with catchup and mayonnaise [28]
“Yautía”—mashed [28]

**Table 2 nutrients-17-00434-t002:** Dietary intake of calories, macronutrients, sodium, potassium and salt (g/day) and 24 h urinary excretion of sodium and potassium (mmol/day) and estimated salt intake (g/day) in the total sample and by sex group.

	Total Sample (n = 69)	Women (n = 36)	Men (n = 33)
	Mean ± SD [Range]	Mean ± SD [Range]	Mean ± SD [Range]
Age (years)	44.4 ± 14.6 [20.0–72.0]	44.8 ± 14.5 [24.0–72.0]	44.1 ± 15.0 [20.0–71.0]
Dietary intake			
Calories	2332.5 ± 669 [1349.0–4218.0]	2056.1 ± 547.7 ^a^ [1349.0–3747.0]	2634.0 ± 665.7 ^b^ [1622.0–4218.0]
Proteins	88.8 ± 28.2 [38.6–230.0]	78.8 ± 19.2 ^a^ [38.6–125.0]	99.8 ± 32.4 ^b^ [53.4–230.0]
Lipids	117.4 ± 56.6 [47.1–345.0]	100.9 ± 48.9 ^a^ [47.1–314.0]	135.4 ± 59.6 ^b^ [63.0–345.0]
Carbohydrates	234.2 ± 86.6 [6.6–510.0]	216.4 ± 77.2 [81.1–419.0]	253.7 ± 93.2 [6.6–510.0]
Sodium intake (mmol/d) Salt intake (g/d)	133.0 ± 59.7 ^A^ [43.0–362.3] 7.7 ± 3.4 ^A^ [2.5–20.8]	130.4 ± 71.9 [43.0–362.3] 7.5 ± 4.1 [2.5–20.8]	135.7 ± 43.7 [89.0–277.0] 7.8 ± 2.5 [5.1–15.9]
Potasium intake (mmol/d)	68.9 ± 21.0 ^C^ [31.0–126.6]	63.3 ±19.4 ^a^ [31.0–119.6]	75.0 ± 21.3 ^b^ [42.2–126.6]
Sodium-to-potassium (mmol/d)	2.0 ± 1.1 [0.8–6.3]	2.1 ± 1.2 [1.0–6.1]	1.9 ± 0.9 [0.8–6.3]
Urine			
Sodium urine (mmol/d)	103.7 ± 44.5 ^B^ [23.0–202.0]	91.8 ± 47.2 ^a^ [23.0–202.0]	116.6 ± 37.9 ^b^ [48.0–192.0]
Estimated salt intake (g/d)	5.9 ± 2.6 ^B^ [1.3–11.6]	5.3 ± 2.7 ^a^ [1.3–11.3]	6.6 ± 2.4 ^b^ [2.3–11.6]
Potassium urine (mmol/d)	36.2 ± 16.3 ^D^ [12.5–84.2]	30.6 ± 15.7 ^a^ [12.5–78.7]	42.2 ± 15.0 ^b^ [14.7–84.2]
Sodium-to-potassium (mmol/d)	3.2 ± 1.6 [1.0–7.9]	3.3 ± 1.8 [1.0–7.9]	3.0 ± 1.3 [1.0–7.6]

Superscripts ^a,b^ indicate differences between sexes and superscripts; ^A—D^ indicate differences between dietary and urinary data.

**Table 3 nutrients-17-00434-t003:** Correlations between dietary Na and K intakes and dietary components, sex and calorie intake. Pearson’s coefficient: r (*p*-value).

	Na Intake	Sex	Calories	Protein	Lipid	Carbohydrates
Na intake			0.360 (0.002)	0.341 (0.004)	0.325 (0.006)	
K intake	0.313 (0.009)	0.293 (0.0015)	0.630 (0.000)	0.608 (0.000)	0.422 (0.000)	0.318 (0.008)

**Table 4 nutrients-17-00434-t004:** Statistically significant results of multivariate regression analysis of macronutrient and calorie intakes and sex, age and BMI in relation to dietary Na and K intakes.

	β (SE)	*p*
Na intake		
Calories	0.000 (0.000)	<0.001
Overweight	0.095 (0.044)	0.036
K intake		
Calories	0.000 (0.000)	0.000
Proteins	0.358 (0.129)	0.007
Lipids	−0.303 (0.119)	0.013

β: beta coefficient, (SE): standard error, *p*-value.

## Data Availability

The original contributions presented in this study are included in the article/Appendix A. Further inquiries can be directed to the corresponding authors.

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
