# Peer review of "Assessment of Dietary Sodium, Potassium and Sodium-Potassium Ratio Intake by 72 h Dietary Recall and Comparison with a 24 h Urinary Sodium and Potassium Excretion in Dominican Adults"

_nutrients, 2025, doi:10.3390/nu17030434_

Round 1
Reviewer 1 Report
Comments and Suggestions for Authors
For specifically a Dominican population, urinary Na and K are compared to estimates from dietary recall to look for correlations between the levels of the two cations determined by each technique and with a variety of health-related variables in this manuscript. The manuscript is well written, easy to read, and presents the results well, and the subject is interesting and appropriate for this journal. I only have minor comments.
The manuscript requires another round of proofing for grammar and typographical errors. For example in the abstract, the sentence on line is actually a sentence fragment, lacking a verb. In line 25, no space between number and unit. In line 35, "these" should be "These". Two compound sentences are not properly punctuated where there are understood verbs.
Be certain, all abbreviation are defined at first use. For example, WHO in line 47.
Avoid the use of first person voice (e.g., "we") except in introduction and conclusion.
The number of places in numbers and their SD's are not often not consistent (e.g., 68.9+/-21).
The manuscript would be strengthened if a brief paragraph on its limitations were added to the discussion section.
Author Response
Reply to REVIEWER 1
Comments and Suggestions for Authors
For specifically a Dominican population, urinary Na and K are compared to estimates from dietary recall to look for correlations between the levels of the two cations determined by each technique and with a variety of health-related variables in this manuscript. The manuscript is well written, easy to read, and presents the results well, and the subject is interesting and appropriate for this journal. I only have minor comments.
The manuscript requires another round of proofing for grammar and typographical errors.
Reply: Changes have been made throughout the manuscript.
For example in the abstract, the sentence on line is actually a sentence fragment, lacking a verb. Reply: A verb was included in the sentence in lines 24-25.
In line 25, no space between number and unit. Reply: This has been corrected.
In line 35, "these" should be "These". Reply: This has been corrected.
Two compound sentences are not properly punctuated where there are understood verbs. Reply: Lines 35-37 have been slightly modified.
Be certain, all abbreviation are defined at first use. For example, WHO in line 47. Reply: Done. The following abbreviations have been defined for the first time: CVD, Na, K, Na-to-K, WHO, AHA,ADA, SICA, NCD, PAHO/WHO, BMI, CONABIOS, FCT, INCAP, USDA.
Avoid the use of first person voice (e.g., "we") except in introduction and conclusion. Reply: Reply: Changes have been made throughout the manuscript, with the exception of the introduction and conclusions.
The number of places in numbers and their SD's are not often not consistent (e.g., 68.9+/-21).
Reply: I assume you are referring to the number of decimal places in each number. In the first version, I removed the decimal point where it was zero. In this new version of the manuscript, I have included it in all the figures where it was missing.
The manuscript would be strengthened if a brief paragraph on its limitations were added to the discussion section.
Reply: A paragraph on the strengths and limitations of this study has been added on lines 703-714: “Relevant strengths of this study are the detailed identification of the sources/recipients of the Dominican cuisine, the careful identification of its composition using appropriate FCT, the emphasis on questions about salt intake (salt added at the table and during cooking, food seasonings, etc.) and the use of a 72-hour food recall, instead of the 24-hour recall generally used in most studies, to better represent the usual consumption patterns of the participants. However, there are also limitations, such as the fact that this sample is a subset of a previous study in which the sample size was calculated to test for differences between normotensive and hypertensive subjects [23], so the comparison between dietary and 24-hour urinary excretion data was performed on a sample size that was not calculated for this purpose. In addition, the Dominican participants are not representative of the Dominican population and therefore the results cannot be extrapolated to a population level.”

Reviewer 2 Report
Comments and Suggestions for Authors
The article addresses the primary question of whether there is a relationship between self-reported sodium and potassium intake and their 24-hour urinary excretion. This topic may be of interest to readers of Nutrients, despite the findings being based on a very specific geographical region.
The manuscript includes all the elements of an original research article.
The introduction is well-written and draws on a comprehensive body of literature, providing a solid foundation for understanding the background and current state of the topic.
The methodology is highly detailed and well-described, enabling replication of the study if the outlined steps are followed scrupulously. It would be valuable to clarify how the sample size was determined and to assess whether 69 participants constitute an adequate sample size. The statistical analysis conducted is appropriate for achieving the stated objectives.
The results section is thorough, supported by tables and figures that facilitate comprehension and interpretation. A potential recommendation would be to include Tables 1, 2, and 3 as supplementary material. Table 6, which presents the results of the multivariate analysis, is somewhat unclear—specifically, the interpretation and relevance of the odds ratios require further explanation.
The discussion is well-structured and relatively extensive, supported by an abundance of references. This allows for a clear comparison of the study’s findings with those of other authors, highlighting areas of agreement or discrepancy. However, the inclusion of a section on the study's strengths and limitations, as well as a paragraph outlining its specific contributions, would enhance the discussion.
The bibliography is extensive but somewhat outdated in the introduction, with an obsolescence index (median publication age) of 8 years. Furthermore, references 25, 29, 30, and particularly reference 31 (from Hola magazine) are derived from non-academic sources, which may detract from the scholarly rigor of the work.
Author Response
Reply to REVIEWER 2
Comments and Suggestions for Authors
The article addresses the primary question of whether there is a relationship between self-reported sodium and potassium intake and their 24-hour urinary excretion. This topic may be of interest to readers of Nutrients, despite the findings being based on a very specific geographical region.
The manuscript includes all the elements of an original research article.
The introduction is well-written and draws on a comprehensive body of literature, providing a solid foundation for understanding the background and current state of the topic.
The methodology is highly detailed and well-described, enabling replication of the study if the outlined steps are followed scrupulously. It would be valuable to clarify how the sample size was determined and to assess whether 69 participants constitute an adequate sample size. The statistical analysis conducted is appropriate for achieving the stated objectives.
- It would be valuable to clarify how the sample size was determined and to assess whether 69 participants constitute an adequate sample size.
Reply: No sample size calculation was performed for this study because the sample was a subset of a previous study (see lines 97-98 of this manuscript) that aimed to estimate salt intake from 24-hour urinary sodium excretion in a group of normotensive and hypertensive Dominican adults (García-Lithgow et al., Nutrients 2023, 15, 3197. https://doi.org/10.3390/nu15143197). Thus, as described in this previous article: ““To assess differences between normotensive and hypertensive subjects, the sample size was calculated on the basis of the urine sodium excretion data (140.5 _ 34.6 y 150.4 _ 38.8 mEq/d, respectively) [39] using the G*Power Program (Universität Düsseldorf, www.gpower.hhu.de/ accessed on 02/03/2021). A sample size of 77 subjects per group was necessary to obtain a difference in urine sodium excretion (10 mEq/d) with 80% power and an alpha error of 0.05.”
The total sample size was 163 subjects, and a complete 3-day foo recall was obtained from 69 participants. The fact that the sample size was not calculated to compare sodium and potassium intakes using two different methodological approaches can be considered a limitation of this study (limitation is now described in lines 701-712 of this revised version of the manuscript). Although the assessment of salt intake by dietary assessment is important per se, as there is no previous information on Dominicans, the accuracy and reliability of the comparison between the two methods would be higher if a sample size calculation had been performed beforehand.
The results section is thorough, supported by tables and figures that facilitate comprehension and interpretation. A potential recommendation would be to include Tables 1, 2, and 3 as supplementary material. Table 6, which presents the results of the multivariate analysis, is somewhat unclear—specifically, the interpretation and relevance of the odds ratios require further explanation.
- A potential recommendation would be to include Tables 1, 2, and 3 as supplementary material.
Reply: Tables 2 and 3 have now been included as supplementary material (Tables S1 and S2). However, we feel that Table 1 should be included in the manuscript as it provides information on the type of recipes and foods consumed by Dominicans, which has not been reported in the scientific literature to date.
- Table 6, which presents the results of the multivariate analysis, is somewhat unclear—specifically, the interpretation and relevance of the odds ratios require further explanation.
Reply: Table 6 (now is Table 4). The Title of table 4 now reads: “Statistically significant results of multivariate regression …..”
Lines 427-432 now reads: “Table 4 shows the statistically significant results of a multivariate regression analysis used to assess the predictive value of calorie, protein, lipid and carbohydrate intake, sex, overweight and age on dietary Na and K intakes. The R2 for Na intake was 0.162 with a significant direct effect of calories and overweight. However, for the K intake, a higher proportion of the variance, 48% (R2 = 0.476) was explained by the direct effect of calorie and protein intake and the inverse effect of lipid intake.”
The figures for the variables related to K intake are slightly different from those in the first version, as they are now presented using the same model (stepwise) that was used in the regression analysis for Na intake.
The discussion is well-structured and relatively extensive, supported by an abundance of references. This allows for a clear comparison of the study’s findings with those of other authors, highlighting areas of agreement or discrepancy. However, the inclusion of a section on the study's strengths and limitations, as well as a paragraph outlining its specific contributions, would enhance the discussion.
- The inclusion of a section on the study's strengths and limitations, as well as a paragraph outlining its specific contributions, would enhance the discussion.
Reply: A paragraph on the strengths and limitations of this study has been added on lines 703-714: “Relevant strengths of this study are the detailed identification of the sources/recipients of the Dominican cuisine, the careful identification of its composition using appropriate FCT, the emphasis on questions about salt intake (salt added at the table and during cooking, food seasonings, etc.) and the use of a 72-hour food recall, instead of the 24-hour recall generally used in most studies, to better represent the usual consumption patterns of the participants. However, there are also limitations, such as the fact that this sample is a subset of a previous study in which the sample size was calculated to test for differences between normotensive and hypertensive subjects [23], so the comparison between dietary and 24-hour urinary excretion data was performed on a sample size that was not calculated for this purpose. In addition, the Dominican participants are not representative of the Dominican population and therefore the results cannot be extrapolated to a population level.”
The following paragraph has also been added to lines 734-739: “These data on sodium, potassium and salt contribute to filling the information gap on salt intake in Dominican subjects, since this assessment of food recalls, together with the previously published study [23], measured by 24-hour urinary excretion, are the only ones available in this population, whose culinary diversity differs from the gastronomy of other Caribbean countries, where several studies have been carried out.”
- The bibliography is extensive but somewhat outdated in the introduction, with an obsolescence index (median publication age) of 8 years.
Reply: To the best of our knowledge, we have included in the introduction the bibliographical references of the most relevant scientific/public health bodies. However, if you feel that we have omitted important references, please let us know and we will include them.
- Furthermore, references 25, 29, 30, and particularly reference 31 (from Hola magazine) are derived from non-academic sources, which may detract from the scholarly rigor of the work.
Reply: I agree that these references are non-academic, but they are necessary (25,29,30) to know the composition and preparation of the recipes used to calculate the dietary intake of the participants. It would also be important if a replication study were carried out in other Dominican groups. However, we could remove them from the bibliography if the editor tells us how to present this information. Instead, reference 31, which refers to a single recipe, has been removed from the references section and the name of the magazine has been added to the corresponding Table 1.

Round 2
Reviewer 2 Report
Comments and Suggestions for Authors
Is ok this version